# Nitric Oxide Mitigates the Deleterious Effects Caused by Infection of *Pseudomonas syringae* pv. *syringae* and Modulates the Carbon Assimilation Process in Sweet Cherry under Water Stress

**DOI:** 10.3390/plants13101361

**Published:** 2024-05-14

**Authors:** Carlos Rubilar-Hernández, Carolina Álvarez-Maldini, Lorena Pizarro, Franco Figueroa, Luis Villalobos-González, Paula Pimentel, Nicola Fiore, Manuel Pinto

**Affiliations:** 1Laboratorio de Inmunidad Vegetal, Instituto de Ciencias Agroalimentarias, Animales y Ambientales, Universidad de O’Higgins, San Fernando 3070000, Chile; carlos.rubilar@uoh.cl (C.R.-H.); lorena.pizarro@uoh.cl (L.P.); franco.figueroa@uoh.cl (F.F.); 2Instituto de Ciencias Agroalimentarias, Animales y Ambientales, Universidad de O’Higgins, San Fernando 3070000, Chile; caalvarez@udec.cl; 3Departamento de Silvicultura, Facultad de Ciencias Forestales, Universidad de Concepción, Concepción 4070374, Chile; 4Centro UOH de Biología de Sistemas Para la Sanidad Vegetal, Universidad de O’Higgins, San Fernando 3070000, Chile; 5Centro de Estudios Avanzados en Fruticultura (CEAF), Rengo 2940000, Chile; lvillalobos@ceaf.cl (L.V.-G.); ppimentel@ceaf.cl (P.P.); 6Departamento de Sanidad Vegetal, Facultad de Ciencias Agronómicas, Universidad de Chile, Santiago 8820808, Chile; nfiore@uchile.cl

**Keywords:** nitric oxide, *Pseudomonas syringae*, stress mitigation, stomatal conductance, net CO_2_ assimilation

## Abstract

Bacterial canker is an important disease of sweet cherry plants mainly caused by *Pseudomonas syringae* pv. *syringae* (Pss). Water deficit profoundly impairs the yield of this crop. Nitric oxide (NO) is a molecule that plays an important role in the plant defense mechanisms. To evaluate the protection exerted by NO against Pss infection under normal or water-restricted conditions, sodium nitroprusside (SNP), a NO donor, was applied to sweet cherry plants cv. Lapins, before they were exposed to Pss infection under normal or water-restricted conditions throughout two seasons. Well-watered plants treated with exogenous NO presented a lower susceptibility to Pss. A lower susceptibility to Pss was also induced in plants by water stress and this effect was increased when water stress was accompanied by exogenous NO. The lower susceptibility to Pss induced either by exogenous NO or water stress was accompanied by a decrease in the internal bacterial population. In well-watered plants, exogenous NO increased the stomatal conductance and the net CO_2_ assimilation. In water-stressed plants, NO induced an increase in the leaf membranes stability and proline content, but not an increase in the CO_2_ assimilation or the stomatal conductance.

## 1. Introduction

Bacterial canker is one of the most damaging diseases affecting sweet cherries worldwide. It is caused mainly by *Pseudomonas syringae* pv. *syringae* (Pss) and *Pseudomonas amygdali* pv. *morsprunorum* (Pam, formerly *Pseudomonas syringae* pv. *morsprunorum*) [1,2]. This disease causes blossom browning, necrosis with orange-brown gummosis in the trunk and branches, round brown lesions in leaves, and dead buds, impairing the yield and viability of juvenile plants during plantation with a potential loss of 75% [3,4,5]. In Chile, Pss is the major cause of the disease in cherry orchards; however, Pam was recently detected in wood cankers [6,7].

The main strategy to control bacterial canker in sweet cherry plants is the use of resistant varieties, cultural practices, and the application of biocides, most of them containing cupric compounds. However, the extensive use of biocides is lowering their effectiveness due to the emergence of Cu-resistant Pss strains, forcing growers to increase the application frequency and doses to maintain the effectiveness. This has caused toxicity in the plants, soil and water contamination, and other negative environmental effects [8,9]. These problems increase the need for new and more sustainable strategies to control the disease, particularly under the new conditions imposed by an ever-changing climate. Among the factors that modulate bacterial infection, the relationship between water availability and disease intensity is perhaps one of the more important. It is well known that an excess of water (heavy rains, high relative humidity, and flooding) enhances bacterial infection; on the other hand, not much information is available about infection under water scarcity. Drought is one of the most damaging environmental stresses affecting the plant water status and important physiological processes such as photosynthesis, which, in turn, alter the growth and yield of crops, including sweet cherry trees [10,11,12]. Indeed, under mild and severe water restriction, the net CO_2_ assimilation, stomatal conductance, and transpiration rate are heavily reduced in different sweet cherry cultivars [13,14,15]. 

In many circumstances in the field, plants are simultaneously exposed to abiotic and biotic stress. The interactions between both types of stress are complex and not well known. For example, in the case of drought stress and bacterial infection, interaction depends on the bacteria strain, plant genotype, and drought intensity [16]. Severe and mild drought elicits different plant response mechanisms [17], leading to altered plant defenses to pathogen infection. Crop losses caused by phytopathogens could be more significant under drought conditions due to reduced plant defense systems. For example, with increasing levels of drought stress, increased susceptibility has been reported for *Vitis vinifera* inoculated with *Xylella fastidiosa* [18]. Severe drought stress has also increased the susceptibility to *Xanthomonas campestris* pv. *musacearum*, *Xanthomonas oryzae* pv. *oryzae*, *Pseudomonas syringae* pv. *tomato*, and *Pseudomonas syringae* pv. *actinidiae*, in apple, rice, *Arabidopsis thaliana* (*Arabidopsis*), and sweet cherry, respectively [19,20,21]. However, in other cases, a moderate water deficit decreases the susceptibility to *Erwinia amylovora* and *Pseudomonas syringae* pv. *tomato* in apple and *Arabidopsis*, respectively [22,23]. Furthermore, the acclimation of plants to moderate drought stress reduced the *in planta* multiplication of these pathogens [24]. Therefore, the susceptibility to a pathogen depends on the water stress intensity (moderate or severe) and likely on the opportunity at which the stress occurs. 

Plant responses to drought and bacterial infection share some key molecular regulatory elements that could influence plant performance. An example of this is nitric oxide (NO) and another, that interacts with NO under stressed conditions, is abscisic acid (ABA). This phytohormone is a key player in the control of the plant response to drought, inducing stomatal closure—the major process controlling CO_2_ assimilation and transpiration rates—playing a significant role in the mitigation of drought harmful effects [25,26,27]. ABA acts upstream of proline accumulation by regulating the expression of genes encoding the key enzymes in proline biosynthesis [28]. Proline acts as a cell osmotic regulator but also as a reactive oxygen species (ROS) scavenger, protecting the plasma membrane integrity [29,30]. ABA has a contrasting role in plant defense because its accumulation and signaling could increase the susceptivity to bacterial and fungal phytopathogens by suppressing the synthesis of salicylic acid (SA), a well-known defense-related phytohormone [31,32,33]. Moreover, in some host/phytopathogenic bacteria interactions, when the infectious process and drought occur simultaneously, ABA levels do not change, while SA and jasmonic acid were increased [34]. However, ABA can be increased by fungal infection [35,36] and high ABA levels correlate with reduced susceptibility to *Phytophthora cinnamomi* and *Botrytis cinerea* in chestnut and tomato, respectively [37,38].

The other key regulatory molecule induced by both stresses is nitric oxide (NO), a gaseous, free radical, endogenous molecule [39,40]. In plants, two biosynthetic pathways have been detected: a reductive route with nitrite reductase as the key player, and an oxidative route with Nitric Oxide Synthase (NOS)-like activity as the main participant, though the enzyme with this activity has not been yet elucidated [41,42]. NO is required to establish a proper hypersensitive response, impairing the spread of bacterial phytopathogens [43,44]. Moreover, host-generated NO modifies bacterial effectors, impairing their suppressive immune effect in plants [45], and exogenous NO application reduces plant susceptibility to phytopathogen infection by increasing the activity of antioxidant enzymes such as ascorbate peroxidase and catalase [46,47]. Application of exogenous NO also induces an increment of several antioxidant activities, alleviating detrimental drought effects on plant growth and development [48,49,50]. Coherently, the exposure of cherry fruits to NO donors also enhances their antioxidant machinery, decreasing the fruit ROS content, and in turn improving their storability [51]. This suggests that NO-dependent signaling could be triggered in this species and affect the plant bacterial susceptibility and drought stress tolerance. However, in sweet cherry plants, there is no evidence of the NO action and their relationship with bacterial susceptibility and drought stress tolerance, either combined or separately. Based on this, we postulate that spraying sodium nitroprusside (SNP), a NO donor, on leaves of sweet cherry plants decreases their susceptibility to *Pseudomonas syringae* pv. *syringae* even under water-restricted conditions. Therefore, this study indicates that NO-triggered signaling is a crucial factor in preventing sweet cherry bacterial canker, particularly under water-restricted conditions. 

## 2. Results

### 2.1. Characterization of the Symptoms Induced by Pss in Leaves and in Lignified Seasonal Branches

Characterization of the symptoms induced by Pss infection in leaf was performed by scoring the evolution of the necrotic area in leaf disk samples inoculated in vitro with 11116_b1 Pss strain [52]. Unlike the mock condition, in inoculated leaf disks, typical necrosis was evident at seven days and ten days post-inoculation (dpi; Figure 1a,b). In the case of seasonal branches inoculated with Pss, they showed different symptoms, such as gum exudation, browning tissue, and the presence of an open wound around the inoculation point (Figure 2). When the bark covering area to the wound was peeled away, only the Pss-infected branches showed an inner crack that started at the inoculation point (Figure 2b). All these symptoms were then considered as characteristics of 11116_b1 Pss infection for these types of plant tissues. 

### 2.2. Effect of Exogenous NO on the Pss Susceptibility of Leaves and Seasonal Branches

The representative image of the inoculated disks shows the effectiveness of the test detecting differences in the necrotic area caused by Pss in leaves treated with different SNP doses at thirty days post-treatment (dpt; Figure 3a). The evolution of the infection in leaf disks from leaves without SNP treatment (Figure 3b) showed that the necrotic area increases rapidly after inoculation, affecting around 43% of the total area of the disk at ten dpi. However, leaf disks from leaves previously treated with 0.2 mM or 0.5 mM SNP doses initially presented a slower increment in the necrotic area, which finally resulted in a significant reduction regarding the control condition at 10 dpi, around 10% and 7% of the necrotic area, respectively (Figure 3b; Appendix A). These values mean that SNP could reduce approximately 83% the necrotic area provoked by Pss infection. Moreover, disks from 0.2 or 0.5 mM SNP-treated leaves showed necrotic areas of 22 and 20% at 10 dpi in four dpt sampled leaves, respectively, and around 26 and 22% in seven dpt sampled leaves, respectively (Figure 3c; Appendix A). This represents almost half of the necrotic area developed by the leaves that did not receive SNP. Concordantly with this, across all the sampling dates, 0.2 and 0.5 mM SNP-treated leaves showed a consistent and significant decrease in the viable bacterial load compared to the control (Figure 4a), which, in turn, significantly positively correlates (r: 0.6023; *p*: 0.0382) with the relative necrotic area provoked by Pss in disk from leaves exposed to SNP (Figure 4b). 

To corroborate whether the decrease in susceptibility to Pss caused by SNP in leaf disks is also induced *in planta*, fully expanded leaves attached to lignified seasonal branches were sprayed with 0.5 mM of SNP or with 0.01% Tween 20^TM^ solution as a control. Four days later, the branches were inoculated by wounding the tissue 15 cm from the branch apex. After sixty days, mock inoculation showed a smaller wound and lower bacterial load than in Pss-inoculated branches, indicating that external contamination or endophytic bacteria were not a relevant factor to alter Pss infection. Also, no significant external symptoms were observed in Pss-infected branches (Figure 5a,b). However, SNP application (0.5 mM) to the leaves and branches strongly decreased the length of the inner crack by nearly 50% (Figure 5c,d). This result corroborates that the SNP application reduces the susceptibility of branch tissues of juvenile sweet cherry plants. This reduction in the susceptibility was again accompanied by a significant reduction in the viable bacterial load (Figure 5e). 

### 2.3. Effect of Exogenous NO on the Pss Susceptibility under Water Stress

In two growing seasons (2021 and 2022), the combined effect of SNP application (0.5 mM) and water stress on the development of Pss infection in leaves was evaluated. In the first season, leaf disks from plants without SNP but under water restriction (−SNP → WS) at five dpt (or three days after water withholding) demonstrated a significant reduction in the necrotic area value regarding that of leaf disks from well-watered plants (−SNP → WW); 16.9 (±12.9)% vs. 34.4 (±29.7)%, respectively (Figure 6a). This equates to an approximately 51% infection reduction. Interestingly, this reduction, caused by a rather moderated water stress (Ψ_md_ = −1.8 MPa, Figure 7b), was not statistically different to that developed by well-watered plants previously treated with SNP (+SNP → WW) (Figure 6a). This shows that the application of exogenous NO produces a similar effect on the reduction in necrotic area to water stress. Moreover, when the application of SNP was combined with water stress (+SNP → WS), the necrotic area presented the highest reduction (77% reduction; Figure 6a) with respect to −SNP → WW treatment. When the stress was stronger (Ψ_md_ = −2.8 MPa, Figure 7b) at 12 dpt (or ten days after water withholding), a necrotic area decrease in −SNP → WS leaves was also observed, but to a lesser extent than those observed earlier. However, the combination of both treatments (+SNP → WS) again produced the highest and significant reduction in necrosis. This reveals a positive interaction between water stress and SNP application for inducing tolerance to Pss infection (Figure 6b). The decrease in necrosis area produced either by exogenous NO or water stress was also accompanied by a significant decrease in the viable bacterial load (Appendix A).

Regarding the 2022 season, plants of the −SNP → WS treatment were under moderate water stress at 25 dpt (Ψ_md_ = −2.0 MPa, Figure 7d), 20 days later than those of the 2021 season. As explained, this was due to the lower substrate desiccation rate caused by the coverage on pots. Under these water stress conditions, −SNP → WS leaves showed a significant reduction in the necrotic area induced by Pss infection compared to −SNP → WW plants (33.6 (±20.6)% vs. 67.6 (±22.0)%, respectively; Figure 6c). A similar reduction in the necrosis area was observed in +SNP → WW leaves (47.6% ± 18.8%; Figure 7c) showing that exogenous NO was still effective at 25 dpt. In both conditions, the viable bacterial load was again significantly reduced compared to −SNP → WW leaves (Appendix A). A combination of SNP application and WS regime produced an even stronger reduction in the necrotic area compared to −SNP → WW leaves (20.4 (±15.5)%; Figure 6c). These results show again that exogenous NO and water stress interact positively to induce tolerance to Pss infection in juvenile plants of sweet cherry. 

### 2.4. Effect of Exogenous NO on the Physiological Behavior of Plants under the Water Stress Conditions

To evaluate how exogenous NO and an imposed water shortage affected the behavior of juvenile sweet cherry plants, different physiological parameters were measured in parallel to measurements of susceptibility to Pss infection (Appendix A). Water stress was gradually induced by the water loss of pots containing the plants. In the 2021 season, while no water loss was registered during the assay in pots with well-watered plants (WW), water loss decreased sharply until six dpt and then steadily until the end of the assay in pots with plants under a water shortage regime (WS) (Figure 7a). This decrease was consistent with the significant decrease in the mid-day leaf water potential (Ψ_md_), which reached −2.1 (±−0.5) MPa at eight dpt and −3.0 (±−0.8) MPa at 16 dpt (Figure 7b). In accordance with previous reports of juvenile sweet cherry plants cv. Lapins [53], these Ψ_md_ values indicate that the plants were under a moderate water stress at the first leaf sampling date but eight days after, when their Ψ_md_ reached −3 MPa, they were under a severe water stress. 

During the 2022 season, using pots covered with a plastic film, the predawn water potential (Ψ_pd_) remained constant at around −0.6 (±−0.1) MPa until 25 dpt in well-watered plants (Figure 7c). This indicates that the substrate was all the time at field water capacity in well-watered conditions. In water-stressed plants, after the first week, Ψ_pd_ decreased significantly, reaching −1.0 (±−0.2) MPa at 25 dpt (Figure 6c). This decrease was accompanied by a significant Ψ_md_ decrease, reaching −2.0 (±−0.2) MPa at 25 dpt (Figure 7d), indicating that, at this time, plants were under moderate water stress [53]. The results of both seasons also show that, independent of the water condition, applications of SNP did not exert any effect either on water loss or on the leaf water potentials, Ψ_pd_ and Ψ_md_ (Figure 7a–d).

During the 2021 growing season under well-watered conditions, a significant stomatal conductance (g_s_) increment was unexpectedly registered at 7 and 16 dpt in +SNP → WW plants compared to values of −SNP → WW plants. These were almost 30% higher than those presented by leaves not previously sprayed with SNP (Figure 8a). This increase, provoked by exogenous NO, was accompanied by a similar increase in the net CO_2_ assimilation rate (A_n_) in +SNP → WW compared to −SNP → WW across both dates: 15.7 ± 3.2 µmol CO_2_ m^−2^ s^−1^ vs. 11.2 ± 1.7 µmol CO_2_ m^−2^ s^−1^ at 7 dpt and 18.1 ± 1.3 µmol CO_2_ m^−2^ s^−1^ vs. 14.4 ± 3.0 µmol CO_2_ m^−2^ s^−1^ at 16 dpt (Figure 8b). 

Under water stress, the stomatal conductance decreased significantly in −SNP → WS plants as the water restriction progressed, reaching 90 mmol H_2_O m^−2^ s^−1^ at 7 dpt and 30 mmol H_2_O m^−2^ s^−1^ at 16 dpt, when the water stress was severe (Ψ_md_ = −3.0 (±−0.8) MPa) (Figure 8a). This g_s_ decrease was accompanied by a decrease in A_n_ which reached near zero at 16 dpt, when the stomata were probably closed (Figure 8b). In +SNP → WS plants, after 7 days of water restriction, the decrease of g_s_ was as abrupt as in −SNP → WS plants; however, in the case of An, the decrease observed was not so abrupt, remaining at 9.2 (±3.2) µmol CO_2_ m^−2^ s^−1^, a value almost twice to that observed in −SNP → WS plants (4.9 (±0.9) µmol CO_2_ m^−2^ s^−1^) (Figure 8b). This positive effect of exogenous NO on A_n_ of plants under moderated stress finally disappeared at 16 dpt when water stress became severe and A_n_ reached near zero. 

During the 2022 season, due to the coverage of pots, differences in g_s_ between water regimes became evident only at 14 dpt (Figure 8c). In well-watered conditions and as in the first season, +SNP → WW plants again showed a significant increase in g_s_ compared to −SNP → WW plants: 281 (±102.9) mmol H_2_O m^−2^ s^−1^ vs. 160.4 (±57.7) mmol H_2_O m^−2^ s^−1^, respectively (Figure 8c). This increment was again accompanied by a significant increment in A_n_ in +SNP → WW compared to −SNP → WW plants: 11.6 (±3.0) µmol CO_2_ m^−2^ s^−1^ vs. 7.8 (±2.5) µmol CO_2_ m^−2^ s^−1^, respectively (Figure 8d). 

At 14 dpt, all water-stressed plants exhibited a similar g_s_ that averaged 60 mmol H_2_O m^−2^ s^−1^ with no statistical differences between −SNP and +SNP plants (66.8 (±34.6) mmol H_2_O m^−2^ s^−1^ vs. 50.1 (±19.2) mmol H_2_O m^−2^ s^−1^, respectively; Figure 8c). With respect to A_n_, a similar trend to the first season was observed, with +SNP → WS plants exhibiting a higher value with respect to −SNP → WS plants, but not statistically significant (Figure 8d). 

The water transpiration rate (E) was clearly affected by the water shortage when compared to the limited effect induced by exogenous NO exposition in both seasons following a similar pattern as observed in stomatal conductance (Appendix A). Moreover, water usage efficiency (WUE) increased in WS plants at 7 dpt independent of SNP application, with no statistical differences at 16 dpt (Appendix A). In the 2022 season, WUE was not affected by WS or SNP treatment (Appendix A).

### 2.5. Effect of NO on the Leaf Membrane Integrity, the Proline Content, and Photosystem II (PSII) Maximum Quantum Yield

To complement the physiological measurements, the effects of the application of SNP on the leaf membrane integrity and proline content were estimated only at the end of the second season, i.e., at 36 dpt (Figure 9a,b). Leaves from −SNP → WS plants presented a significant increase in the leaf electrolyte leakage with respect to leaves from well-watered plants with or without SNP application (Figure 9a). Interestingly, in +SNP → WS plants, the electrolyte leakage remained at a similar value to that observed in −SNP → WW plants (Figure 9a). This suggests that water stress effectively damages the leaf membranes while exogenous NO exerts a protective effect on membrane integrity. Additionally, water-stressed plants exhibited a significant increase in their leaf proline content (Figure 9b). Interestingly, only under water stress conditions did the SNP-treated plants (+SNP → WS) show a significant increase in the content of leaf proline. This content almost duplicated that exhibited by leaves of −SNP → WS plants (Figure 9b) and was around eight times higher than the leaf contents exhibited by non-SNP-treated plants, regardless of whether they were under water stress. Regarding the PSII maximum quantum yield, it was significantly affected by the water stress regime but not by the SNP application (Figure 9c). 

## 3. Discussion

The necrotic area developed by leaf disks after 10 days from the inoculation with the 11116_b1 Pss strain (around 50%) revealed that this period was adequate to find a high difference with the necrotic area developed by mock leaf disks (around 2%, Figure 1b). This means that from the total response found, almost 96% was due to the Pss infection. Similar results were found after 60 dpi in lignified seasonal branches (Figure 2a,b). Therefore, this result indicates that the endophytic bacteria or external bacterial contamination in the samples after 10 dpi in the case of leaf disks or 60 dpi in the case of branches were not able to develop a significant necrotic area in leaf disks or bark, as the Pss infection did. When the NO donor SNP was applied to leaves previously to this Pss infection, a significant decrease in the leaf disks necrotic area was provoked by 0.2 and 0.5 mM NO at 7 and 30 dpt (Figure 3b,c); according with previous reports, this would be probably due to their antioxidant machinery being enhanced by increasing the activity of antioxidant enzymes [46,47], thus decreasing the reactive oxygen species (ROS) content in the plant. The plant antioxidant machinery plays an essential role in controlling the burst of ROS, ensuring plant tolerance to abiotic and biotic stresses such as drought and phytopathogenic infection [18,54,55]. Among the molecules that are part of this machinery, nitric oxide (NO) participates in effector-triggered immune signaling protection and regulates the activity of plant respiratory burst oxidase homologs (RBOHs) under bacterial infection, thus influencing salicylic acid (SA) signaling [44,45,56]. Foliar application of NO donors like sodium nitroprusside (SNP) can increase the internal content of NO in plant tissues at even one hour after application [57,58]. In accordance with this, the results of the present study demonstrate that application of SNP effectively reduces the susceptibility to *Pseudomonas syringae* pv. *syringae* (Pss) of juvenile sweet cherry plants cv. Lapins (Figure 3 and Figure 5). In addition, SNP doses used in this study were similar to those used in other reports where SNP application, prior to inoculation, decreased the susceptibility to fungal and viral phytopathogens of different plant species [46,59,60]. The protective effect of the exogenous NO in the susceptibility to fungal phytopathogen has been also observed in fruits such as peach and mango [47,61]. This suggests that the effect of exogenous NO application on phytopathogen susceptibility could be a conserved character among plant species, including sweet cherry. 

Different SNP doses sprayed on sweet cherry plants under well-watered conditions decreased their susceptibility to Pss, presenting a significant reduction even 30 days after the application of 0.5 mM SNP. This long-lasting effect of exogenous NO was also observed in lignified seasonal branches where the effects of SNP application prior to inoculation were observed even sixty days after inoculation (Figure 3 and Figure 5). A long-lasting SNP-dependent effect has also been reported to be induced by a pre-harvest application, influencing the post-harvest quality of peach fruit [62] and during fungal infection in muskmelon fruit [63]. The present study expands on this protective effect over bacterial infection in non-reproductive aerial organs that could reply in other crops. This long-lasting effect also suggests that SNP application induced a primed state in juvenile sweet cherry plants, boosting their immune system and promoting a better and more effective response to subsequent phytopathogen infections [64]. However, it was observed that SNP application also impairs Pss population (Figure 4 and Figure 5e and Appendix A). The regulation of defense-related genes and phytohormones should be assessed to corroborate this primed state effect induced by exogenous NO in sweet cherry or other species. A reduction in fungal and bacterial populations in tomato and *Solanum quitoense* has also been attributed to exogenous NO [46,65]. 

Interestingly, like exogenous NO, water stress also reduced the Pss population (Appendix A). This effect on the bacterial population was produced in the first season wherein plants were under moderate water stress (Ψ_md_ = −2.1 MPa) and under severe water stress (Ψ_md_ = −3.0 MPa) at 5 and 12 dpt, respectively (Figure 6a,b). In the 2022 season, this was again observed when the Ψ_md_ value reached −2.0 MPa at 25 dpt (Figure 6c). At the same time, this moderate water stress induced a significant reduction in the susceptibility to Pss (Figure 6a,c). This reduction was similar to that induced in well-watered plants by the SNP application. These results also suggest that a reduction in the susceptibility to Pss, particularly that induced by water stress, can be partially attributed to a reduction in the bacterial population (Appendix A). It is well known that most plant pathogens use stomatal pores as entry points to invade the leaf apoplast and that stomatal closure is also a defense mechanism used by plants [66]. In the case of water-stressed plants, the reduction in the bacterial population correlates well with the reduction in stomatal conductance (Figure 8a,c, Appendix A). Therefore, under water stress, stomata closure could be part of the mechanisms by which water stress reduces the susceptibility to Pss infection in sweet cherry cv. Lapins. However, this study clearly demonstrated in both seasons, that in well-watered plants, exogenous NO increased the stomatal conductance while decreasing the bacterial population and susceptibility to Pss infection. In this case it is possible that the NO overaccumulation induced by exogenous NO [57,58] participates actively in many plant species where its homeostasis is required for its proper defense [67,68], including members of the *Prunus* genus such as peach [61,62]. The NO internal level is controlled by reductive and oxidative enzymatic routes where the NO turnover could be regulated by S-nitrosoglutathione reductase [42,69]. Therefore, both processes involving in NO homeostasis could be participating in the immune response of sweet cherry plants against bacterial phytopathogen such as Pss. In this context, different proteins involved in SA signaling and participating in immune response such as NPR1 and SRG1 can be activated by an NO overaccumulation via S-nitrosylation, [56,70,71]. This post-translational modification also impacts SnRK2.6/OST1, a key player inhibiting ABA signaling and impairing the stomatal closure [72]. Similarly, loss-of-function mutants in ABA biosynthesis and signaling presented a higher g_s_ in *Arabidopsis* and tomato plants, always in non-stressed conditions [73,74]. These results agree with the present study, where under well-watered conditions, exogenous NO induced a significant increment of g_s_ in both seasons (Figure 8a,c). Therefore, after SNP application, the expected internal NO overaccumulation, including guard cells, could impair ABA synthesis and signaling, directly impacting on the stomatal opening. This increment in g_s_ and A_n_ induced by exogenous NO is consistent with a previous report in well-watered watermelon plants [75]. In several plant species, increased g_s_ normally allows an increase in CO_2_ diffusion into the leaf, which, in turn, results in a higher net CO_2_ assimilation rate [76]. In concordance with this, and under well-watered conditions, the increased g_s_ observed in the present study was followed by a significant increment in the net CO_2_ assimilation rate of SNP-treated plants (Figure 8b,d). However, this increase in g_s_ was not clearly followed by an increase in the water transpiration rate (E). Only in the second season did SNP produce a significant increment in E at 16 days after its application in well-watered plants. In fact, the E and water usage efficiency were clearly more affected by the water restriction than by the SNP application.

Under water stress conditions, a positive effect caused by SNP application on A_n_ was also observed at 7 dpt in 2021 season. These results agreed with previous report that indicate that under salt stress SNP applications improve A_n_ and g_s_ in barley [77]. However, in our case, the A_n_ improvement was not accompanied by an improvement of g_s_, as was observed in well-watered conditions in both seasons (Figure 8b). This suggests that, at this date, g_s_ was not modulated by exogenous NO, probably because it was not capable of mitigating the water-stress-dependent stomatal closure. Further, this also suggests that, under water stress, the high A_n_ value caused by the exogenous NO was not totally dependent on the modulation of stomatal conductance and other factors played a role in modulating A_n_. A hypothesis to explain this could be the protection that exogenous NO exerted on leaf cell membrane integrity (Figure 9a). The positive effect of exogenous NO on the membrane stability under water stress conditions has been reported [49,50], and it has also been reported that exogenous NO mitigated PSII photochemistry impairment induced by heat stress in rice [78], but this NO-dependent mitigation of PSII damage seems in turn be dependent on the NO-donor concentration and on the plant species [79]. In this study, there was not a clear mitigation of the exogenous NO on the PSII damage caused by water stress (Figure 9c), suggesting that thylakoid membrane stability probably was not playing an important role in the positive effect caused by SNP application on A_n_. This also suggests that, in this case, exogenous NO effectively decreases harmful water stress effects, specifically mitigating instability of cell membranes other than those of thylakoids. Another possibility could be changes in processes that are part of the plant carbon balance, such as photorespiration and dark respiration [76,80]. For example, previous reports indicated that the damage caused in plants by toxic metals like As and Cd were attenuated by exogenous NO, which, in turn, produced a decrease in photorespiration but an increase in the dark respiration rate throughout the activation of the alternative pathway [81,82]. Taken together, this study shows that, in plants with an optimal water status, exogenous NO improves the stomatal conductance and net CO_2_ assimilation; however, under water stress, this positive effect on A_n_ needs to be more investigated because it was evident only at moderated water stress in the first season, not being statically significant in the second one.

As already shown, water stress could contribute to decreased susceptibility to Pss infection by closing the stomata (Figure 8) and affecting the viable bacterial load (Appendix A). However, it could also be involved in inducing many other mechanisms that contribute to plant protection. For example, promoting callose deposition next to the sites of pathogen penetration, which is an important factor for plant defense against invading pathogens [83,84]. Moreover, the transcription of pathogen-related genes such as thaumatin-like protein genes is activated in pathogens by NO/ABA signaling, participating in drought responses such as stomatal closure, and the accumulation of osmolytes such as proline [85]. In fact, water stress induces the accumulation of this osmolyte [28,29,30], proline metabolism contributes to defense responses in plants facing phytopathogen infection [86], and proline accumulation mitigates lower water potential effect to sustain cell turgor by osmotic adjustment [87,88]. Interestingly, exogenous NO also induced the accumulation of proline in plants under another stress such as high salinity [89,90]. In the present study, the proline content was significantly increased in leaves by water stress but, curiously, not by exogenous NO itself. However, when both treatments were applied together, the proline content increased synergistically, doubling the value of water-stressed leaves and being almost eight times higher than the values of leaves from well-watered plants receiving exogenous NO (Figure 9b). Therefore, water restriction induced the accumulation of this osmoprotectant, exerting protection against both plant pathogenic bacteria infection and drought deleterious effects in sweet cherry plants which was enhanced by exogenous NO. 

## 4. Materials and Methods

### 4.1. Plant Material and Growth Conditions

Two-year-old cv. Lapins sweet cherry plants, grafted on Maxma 14 rootstock, were obtained from a commercial plant nursery (lat.: −34.47°; long: −70.98°) early in the spring of the 2021 and 2022 growing seasons. During both seasons, plants were transplanted into 20 L plastic pots filled with a mixture of peat moss, compost, perlite, and soil (3:3:3:1, *v*/*v*/*v*/*v*) as a substrate and fertilized with 3 g L^−1^ substrate of Basacote^®^ plus commercial fertilizer. After transplanting, plants were acclimated for six weeks in a side-open shade house at the Universidad de O’Higgins Experimental Station, San Fernando, Chile (lat: −34.61°, long: −70.99°). During this post-transplanting period, plants were drip irrigated twice per day, 45 min each with a flow of 1.13 L h^−1^. In both seasons, the environmental conditions in the side-open shade house were recorded using a multiple sensor system connected to a data logger (models VP4meter and ZL6, METER Group, Pullman, WA, USA) (Appendix A; Appendix A).

### 4.2. SNP Treatments and Leaf Sampling

Transplanted plants were profusely sprayed with 80 mL plant^−1^ of 0.05, 0.2, and 0.5 mM sodium nitroprusside (SNP, Sigma-Aldrich n° 71778, St. Louis, MO, USA) with 0.01% Tween 20^TM^ as a surfactant–adjuvant agent or only with 0.01% Tween20^TM^ in a control condition. At 4, 7, and 30 days post SNP treatment (dpt), fully expanded leaves were removed from the plants and immediately transported to the laboratory to perform in vitro Pss infection assays.

### 4.3. Bacterial Inoculum

All the inoculations were performed with the Pss strain N° 11116_b1, which was isolated from sweet cherry orchards in the Central Valley of Chile [52]. Pss suspension was grown for 20 h at 26 °C on King’s B medium (KB) agar medium and resuspended in the saline buffer with a concentration adjusted to 2 × 10^8^ colony-forming units (CFU)/mL with 0.01% Tween20^TM^. 

### 4.4. In Vitro Pss Infection Assay

For in vitro assays, a leaf disk infection method was performed [91]. Briefly, fully expanded leaves were washed with sterile distilled water and disinfected with 70% ethanol. Thereafter, leaf disks (1.5 cm of diameter) were obtained using a cork borer previously immersed for 10 s in a Pss suspension containing 2 × 10^8^ CFU/mL with 0.01% Tween 20^TM^. Leaf disks were placed in a Petri plate with the adaxial face touching the 0.8% agar medium. Pictures of the disks were taken at 4, 7, and 10 days post Pss inoculation with a similar brightness and distance from an EOS Rebel T100 camera (Canon, Tokyo, Japan) on a tripod. For each disk, the loss of green area leading to browning was analyzed using FIJI (https://fiji.sc/) [92]. In the green channel, the non-necrotic area in a disk was calculated using a “3D object counter” tool measuring the sum of the area of each generated object above 200 pixels with a threshold around 120. The necrotic area was then calculated from the difference between the measured non-necrotic area and the whole disk area. This difference was relativized by the whole disk area.

### 4.5. In Planta Pss Infection Assay

The *in planta* infection assays were performed in lignified seasonal branches [4] from plants treated four days before with 0.5 mM SNP. Briefly, branches were disinfected with 70% ethanol and were deeply wounded at 15 cm from the apex (between the 4th and 5th apical leaves) with a sterile scalpel. The wound was immediately inoculated with 50 µL of the Pss suspension (2 × 10^8^ CFU/mL) covered with a sterile glycerol solution and immediately sealed with a wax film (Parafilm^®^ M) to avoid dehydration. Two months later, the disease severity (DS) in a branch was assessed by evaluating the presence of the symptoms of browning, open wound, and gum secretion using the following equation: (DS = [(number of found symptoms in a branch)/3] × 100. Also, after peeling the inoculated zone, the size of the inner injury induced by the infection was measured from the inoculation point to the top of the observed cracking with a hand ruler. 

### 4.6. Bacterial Counting

At the end of Pss infection assays, inoculated tissue—a leaf disk or a branch sample near the inoculation site—were sampled and homogenized in a sterile saline buffer (0.136 M NaCl, 0.086 M NaH_2_PO_4_, 0.014 M Na_2_HPO_4_). A dilution series was performed and three 10 µL drops from three different dilution series were placed on KB agar and incubated for 20 h at 26 °C. The colonies with a *Pseudomonas syringae* morphology were counted to determine the viable bacterial load as the number of colony-forming units (CFU) per disk area or per gram of inoculated branch.

### 4.7. Water Restriction Assays

Two water restriction assays were performed during two different growing seasons: October–November 2021 and October–November 2022. During both seasons, six weeks after transplanting to plastic pots, plants were treated with 0.5 mM SNP (+SNP) or only with 0.01% Tween20^TM^ solution as surfactant–adjuvant agent (−SNP). Two days later (2 dpt), plants were split in two halves, and the water supply was withheld in one half to allow a progressive water shortage (WS). In the other half, plants were drip irrigated (WW) twice per day, 45 min each, with a flow of 1.13 L h^−1^. Therefore, the treatments were as follows: (a) 0.01% Tween 20^TM^/well-watered plants (−SNP → WW); (b) 0.5 mM SNP/well-watered plants (+SNP → WW); (c) 0.01% Tween 20^TM^/water shortage plants (−SNP → WS) and (d) 0.5 mM SNP/water shortage plants (+SNP → WS) (Appendix A).

In the 2021 season, the assay was maintained for 22 days, and pots with plants were kept open to favor a rapid substrate desiccation. Water loss was recorded by weighing each pot (W) plus substrate and plant at water field capacity, two days before SNP treatment (W_−2_) and at 0, 6, 9, and 22 days (W_day_) after SNP treatment. Water loss (WL) from each pot was calculated as follows: WL = W_day_ − W_−2_. Each treatment was composed of four experimental units, with three plants per unit, giving 12 plants per treatment, i.e., 48 plants in total (Appendix A). In the 2022 season, the assay was maintained for 36 days and pots were covered with a plastic cover to favor a slow substrate desiccation. In this season, instead of the pot weighing, which is highly time consuming, the pre-dawn water potential (Ψ_pd_) was used as a proxy for the soil water content [93]. Each treatment was composed of three experimental units, with three plants per unit, giving 9 plants per treatment, i.e., 36 plants in total (Appendix A). In both seasons, experimental units were disposed randomly according to their water treatments.

### 4.8. Leaf Water Potential and Gas Exchange Measurements

Different physiological parameters were measured throughout the water restriction assays (Appendix A). The mid-day water potential (Ψ_md_) was measured in detached leaves from the upper third of the plant (six plants per treatment) using a Scholander model 1505D-EXP pressure chamber (PMS Instruments, Albany, OR, USA) [94]. Leaves were covered with aluminum foil before detachment and Ψ_md_ measured at 0, 8, and 16 dpt. In the 2022 season, measurements were performed at 0, 6, and 25 dpt, and measurements of the pre-dawn water potential (Ψ_pd_) were included.

Gas exchange was performed on one leaf per plant placed in the upper third of the plant using a photosynthetic chamber connected to a portable infrared gas analyzer (model CIRAS-3, PP Systems, Amesbury, MA, USA). The CO_2_ concentration in the chamber was adjusted to 400 ppm, the leaf temperature was maintained at 25 ± 1 °C, and the photosynthetically active radiation (PAR) was set to 800 µmol photons m^−2^ s^−1^. Leaves were acclimated to the chamber conditions for at least thirty seconds before measurements. The stomatal conductance (g_s_), net CO_2_ assimilation (A_n_), and transpiration rate (E) were obtained. Instantaneous water use efficiency was calculated as the ratio between A_n_ and E. The measurements were performed at 0, 7, and 16 dpt in the 2021 season and at 0, 6, and 14 dpt in the 2022 season. Time 0 dpt corresponds to the moment when SNP was applied. In both seasons, 8 plants per treatment were assessed for water loss, water potential, and gas exchange measurements unless otherwise specified. All the measurements—except Ψ_pd_—were performed at 13 PM ± 2 h local time.

### 4.9. Maximum Quantum Yield Efficiency, Electrolyte Leakage, and Proline Content

Additional physiological measurements (the maximum quantum yield efficiency, electrolyte leakage, and proline content) were performed only during the 2022 season (Appendix A). The maximum quantum yield efficiency of PSII (Fv/Fm) was measured on one attached leaf per plant (two plants per experimental unit, six plants per treatment) using a pulse-amplitude modulated fluorimeter (FMS2, Hansatech Instruments, Norfolk, England). Leaves were darkened with aluminum foil for 20 min before measurements. Minimal fluorescence (Fo) was obtained by applying a weak modulated light pulse (0.4 µmol m^−2^ s^−1^), and maximal fluorescence (Fm) was induced with a 0.8 s light saturating pulse (9000 µmol PAR m^−2^ s^−1^) [95]. Measurements were performed at 6, 10, 17, 25, and 36 days after SNP treatment. 

Electrolyte leakage (EL) was assessed from three plants per experimental unit (two measurements per experimental unit, six times per treatment). Three leaves per plant were briefly washed with water and cut into 1 cm diameter disks with a cork borer: four disks were cut per leaf and then split in two groups (18 disks in total per measurement). The disks were incubated in orbital rotation, in 20 mL distilled water at 22 °C ± 1 °C, and the conductivity of the resulting solution (C) was recorded at 15 min and 24 h of incubation. After that, the disks were autoclaved, and the conductivity was again measured. The calculation of EL was performed using the following formula: EL = [(C_24h_ − C_15min_)/(C_autoclaving_ − C_15min_)] × 100 [96]. The EL measurements were performed at 36 dpt using an electrolytic conductivity meter (STARTER 3100M bench meter, Ohaus Corporation, Parsippany-Troy Hills, NJ, USA).

The proline content was assessed from a composite sample of three plants per experimental unit (three samples per treatment). Frozen leaf tissue weighing 0.35 g was extracted with 3% sulfosalicylic acid using a batch mill (IKA A11, Staufen, Germany). Afterward, the samples were centrifuged at 4000× *g* for 20 min at 4 °C. Subsequently, a volume of the supernatant was combined with a volume of glacial acetic acid and a volume of the ninhydrin reagent (2.5% ninhydrin [2,2-dihydroxyindane-1,3-dione] in a glacial acetic acid/6 M orthophosphoric acid solution (3:2)). Then, samples were incubated at 100 °C for one hour in a water bath and later cooled and centrifuged for 5 min at 2000× *g* at 4 °C. The supernatant was measured at 520 nm using a spectrophotometer (Jenway 6405 UV/Vis, Chicago, IL, USA) [97]. The proline content was obtained using a standard curve made from commercial p.a. proline. The proline content was performed at 36 dpt.

### 4.10. Statistical Analysis

All the results in the figures were expressed as mean ± SEM and as mean ± SD in text. Data were analyzed using GraphPad Prism version 8.0 for Windows (San Diego, CA, USA, www.graphpad.com). Statistical differences between means were evaluated using a one-way ANOVA or two-way ANOVA followed by Tukey’s or Sidak’s multiple comparisons test (*p* < 0.05 was considered statistically significant). If the data were non-parametrical, a Mann–Whitney U test was performed. In a temporal course assay, a repeated measures two-way ANOVA followed by Tukey’s multiple comparisons test was performed. A correlation test using the Pearson correlation coefficient with a significantly non-null correlation hypothesis was performed.

## 5. Conclusions

From the results obtained in this study, it is possible to conclude that foliar applications of SNP sprayed on juvenile sweet cherry plants prior to Pss inoculation, and under well-watered conditions, effectively reduce the susceptibility to Pss infection. This reduction was still significant in leaves 30 days after SNP application and after 60 days in lignified seasonal branches. This long-lasting effect suggests that exogenous NO induces a primed state in juvenile plants, preparing their immune system for subsequent phytopathogenic infections. The results also indicate that moderate and severe water stress also decrease the susceptibility to Pss infection, and this reduction is increased by exogenous NO applications, denoting a positive interaction between both treatments. This was coherent with the mitigating NO effect on membrane stability and an increased proline content exerted by exogenous NO in a water-restricted condition. It was also demonstrated that water stress and SNP application independently impaired the Pss population, suggesting that a reduction in the susceptibility to Pss caused by these factors could be partially due to this effect. On the other hand, this study clearly demonstrated that exogenous NO increased the stomatal conductance in well-watered plants, producing in turn a significant increment in the net CO_2_ assimilation rate. This positive effect was also observed during the first season at 7 days after SNP application in moderated water-stressed plants. However, as this CO_2_ assimilation increment was observed only in one season and was not linked to an increase of the stomatal conductance, this finding needs to be further investigated. 

## Figures and Tables

**Figure 1 plants-13-01361-f001:**
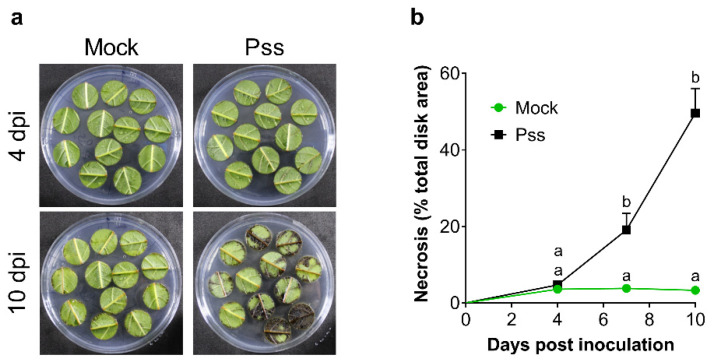
Characterization of Pss infection in sweet cherry leaves. Sweet cherry cv. Lapins fully expanded leaves were inoculated in vitro with Pss (2 × 10^8^ CFU/mL) or mock (0.01% Tween 20^TM^). Representative images of 4- and 10-day post-inoculation (dpi) leaf disks (**a**) and the temporal course of the relative necrotic area in the disks in both conditions (**b**). The numbers of leaf disks used were 12 in mock and 20 in Pss inoculation. Graph represents the mean + SEM. Different letters between two means at the same time indicate statistically significant differences in a two-way ANOVA with Sidak multiple comparisons test (*p* < 0.05).

**Figure 2 plants-13-01361-f002:**
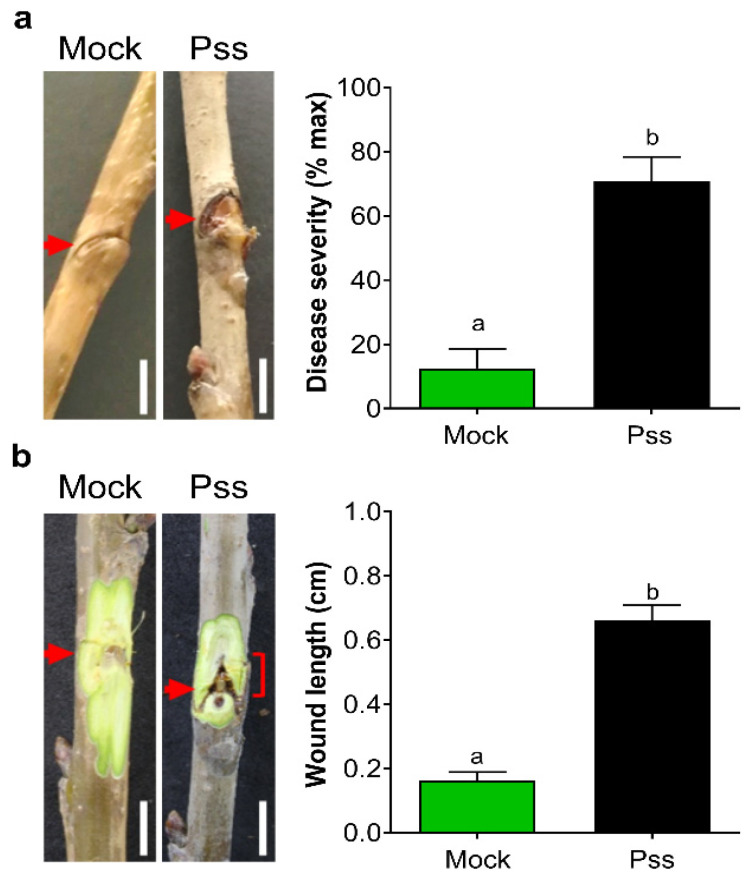
Characterization of Pss infection in sweet cherry branches. Sweet cherry cv. Lapins seasonal lignified branches were inoculated with Pss (2 × 10^8^ CFU/mL) or mock (0.01% Tween 20^TM^). Branches with (**a**) and without bark (**b**) after 60 days post-inoculation. An arrow indicates the point where the branch was inoculated, and the square bracket demarcates the length of the inner wound. The number of branches used in both mock and Pss inoculation was 8. Graphs represent the mean with SEM. Different letters indicate statistically significant differences in an unpaired two-tailed Mann–Whitney U test (*p* < 0.05). Scale bar, 1 cm.

**Figure 3 plants-13-01361-f003:**
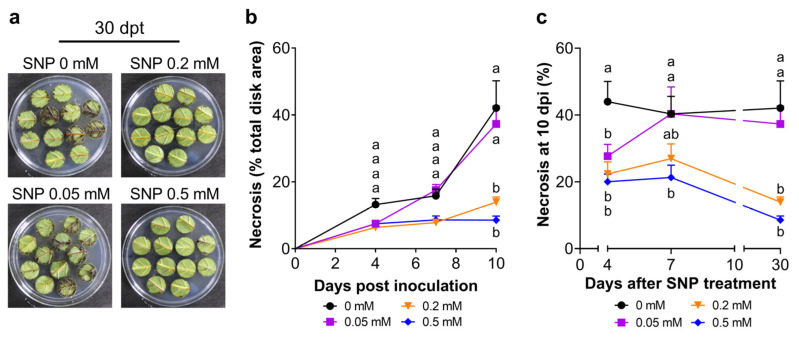
Evolution of Pss infection in sweet cherry leaves exposed to exogenous nitric oxide. Sweet cherry cv. Lapins leaves were treated with 0, 0.05, 0.2, and 0.5 mM SNP and inoculated with Pss (2 × 10^8^ CFU/mL). (**a**) Representative images of 10-day post-inoculation (dpi) leaf disks from 30-day post-treatment (dpt) samples. (**b**) Temporal course of the relative necrotic area in these disks in each condition. (**c**) Quantification of the relative necrotic area of 10 dpi disks from 4, 7, and 30 dpt samples in each SNP condition. (**b**) n = 12 disks; (**c**) n = 32 disks in 4 and 7 dpt samples; 12 disks in 30 dpt samples. Graphs represent the mean with SEM. Different letters between means at the same time indicate statistically significant differences in a two-way ANOVA with Tukey’s multiple comparisons test (*p* < 0.05).

**Figure 4 plants-13-01361-f004:**
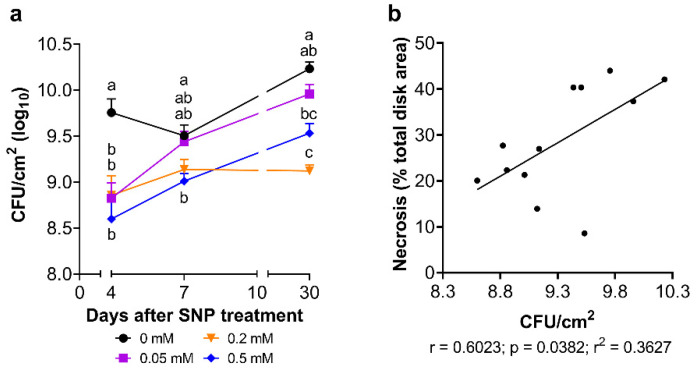
Effect of exogenous nitric oxide on the viable bacterial load in Pss-inoculated sweet cherry leaves. Sweet cherry cv. Lapins leaves were treated with 0, 0.05, 0.2, and 0.5 mM SNP and inoculated with Pss (2 × 10^8^ CFU/mL). (**a**) Quantification of the viable bacterial load in 10-day post-inoculation disks from 4-, 7- and 30-day post-treatment (dpt) samples in each SNP condition (n = 18 in 4 and 7 dpt; n = 9 in 30 dpt). Graph represents the mean with SEM. Different letters between means at the same time indicate statistically significant differences in a two-way ANOVA with Tukey’s multiple comparisons test (*p* < 0.05). (**b**) Relationship between the viable bacterial load and the relative necrotic area in leaf disks. Pearson correlation coefficient (r), the significance to non-zero correlation (*p*), and its coefficient of determination (r^2^) are presented.

**Figure 5 plants-13-01361-f005:**
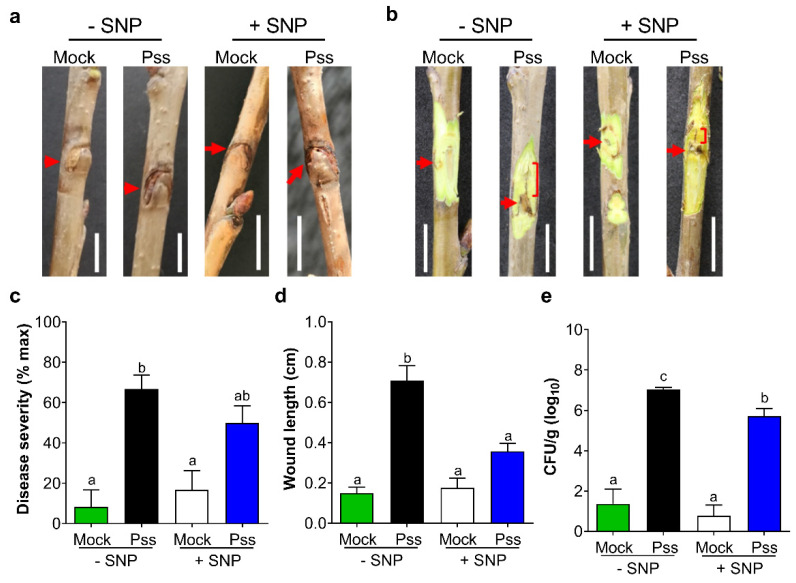
Effect of exogenous nitric oxide in Pss infection on seasonal branches. Sweet cherry cv. Lapins seasonal branches were treated with 0 mM (−SNP) or 0.5 mM (+SNP) SNP and inoculated with Pss (2 × 10^8^ CFU/mL) or mock (0.01% Tween 20^TM^). Seasonal branches with (**a**) and without bark (**b**) after 60 days post-inoculation. An arrow indicates the point where the branch was inoculated, and the square bracket demarcates the length of the necrosis. (**c**) Quantification of the disease severity in branches. (**d**) The length of the necrosis and (**e**) the viable bacterial load next to the wound. (**c**,**d**) n = 4 in mock; 12 in Pss. (**e**) n = 12 in mock; 36 in Pss. Graphs represent the mean with SEM. Different letters indicate statistically significant differences in a two-way ANOVA with Tukey’s multiple comparisons test (*p* < 0.05). Scale bar, 1 cm.

**Figure 6 plants-13-01361-f006:**
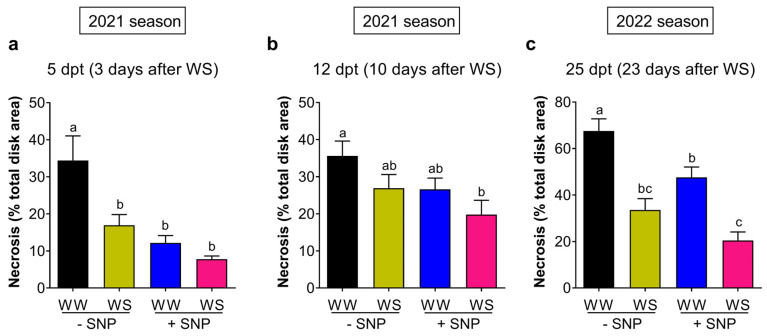
Effect of exogenous nitric oxide on the sweet cherry susceptibility to Pss under water restriction. Sweet cherry cv. Lapins plants were treated with 0 mM (−SNP) or 0.5 mM (+SNP) SNP. After two days, plants maintained their irrigation (WW) or were exposed to a complete water shortage (WS) during two independent seasons. After that, sampled leaves were inoculated with Pss (2 × 10^8^ CFU/mL). Relative necrotic area in leaf disks from leaves sampled in the first season at (**a**) 5 days and (**b**) 12 days after the SNP-treatment leaf samples, and (**c**) 25 days after SNP treatment in the second season. (**a**,**b**) n = 20; (**c**) n = 18. Different letters indicate statistically significant differences in a two-way ANOVA with Tukey’s multiple comparisons test (*p* < 0.05).

**Figure 7 plants-13-01361-f007:**
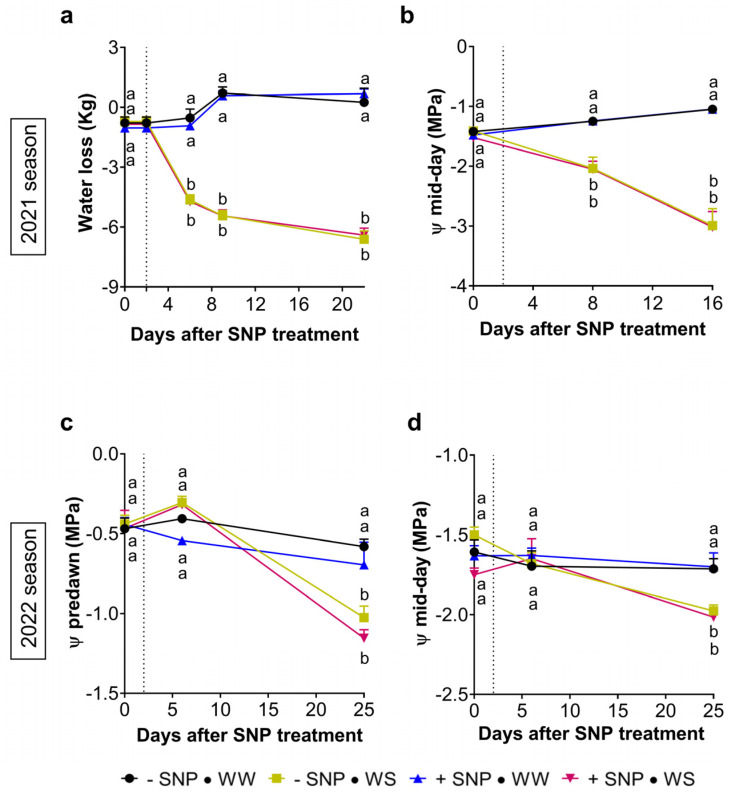
Effect of NO on the evolution of pot water loss and leaf water potential in sweet cherry exposed to exogenous nitric oxide under water restriction. Sweet cherry cv. Lapins plants were treated with 0 mM (−SNP) or 0.5 mM (+SNP) SNP. After two days, plants maintained their irrigation (WW) or were exposed to a complete water shortage (WS, denoted by a dashed vertical line) during two independent seasons. (**a**) Temporal course of changes in water loss from two days before SNP treatment and (**b**) the leaf mid-day water potential (Ψ_md_) in each condition during the water restriction assay in the first season. (**c**) Temporal course of the leaf pre-dawn water potential (Ψ_pd_) and (**d**) the mid-day water potential during the assay in the second season. (**a**,**b**) n = 8; (**c**,**d**) n = 6. Different letters between means at the same time indicate statistically significant differences in a two-way ANOVA with Tukey’s multiple comparisons test (*p* < 0.05).

**Figure 8 plants-13-01361-f008:**
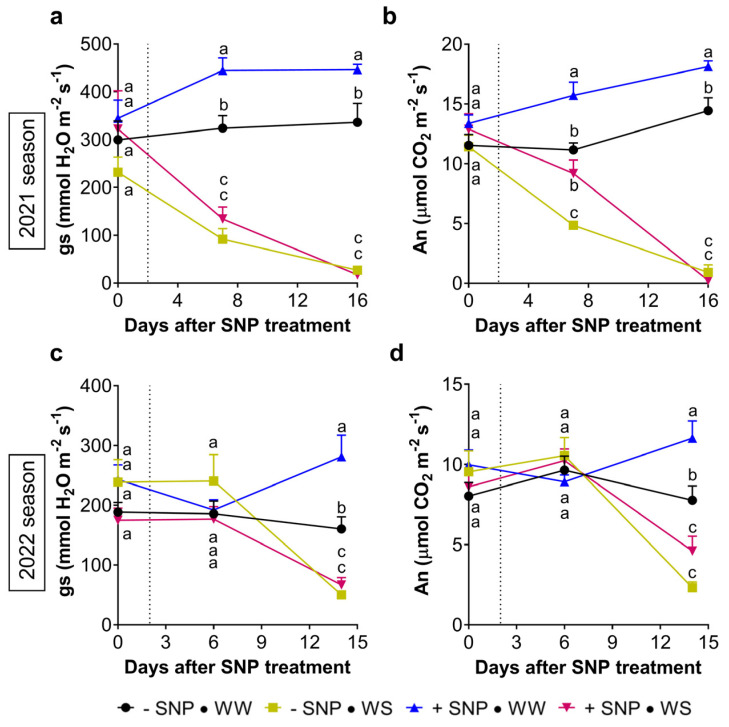
Effect of exogenous NO and water restriction on evolution of stomatal conductance and net CO_2_ assimilation in sweet cherry plants. Sweet cherry cv. Lapins plants were treated with 0 mM (−SNP) or 0.5 mM (+SNP) SNP. After two days, plants maintained their irrigation (WW) or were exposed to a complete water shortage (WS, denoted by a dashed vertical line) during two independent seasons. Temporal course of (**a**,**c**) the stomatal conductance (g_s_) and (**b**,**d**) the net CO_2_ assimilation (A_n_), during the water restriction assay in the first (**a**,**b**) and the second season (**c**,**d**), respectively. n = 8. Different letters between means at the same time indicate statistically significant differences in a two-way ANOVA with Tukey’s multiple comparisons test (*p* < 0.05).

**Figure 9 plants-13-01361-f009:**
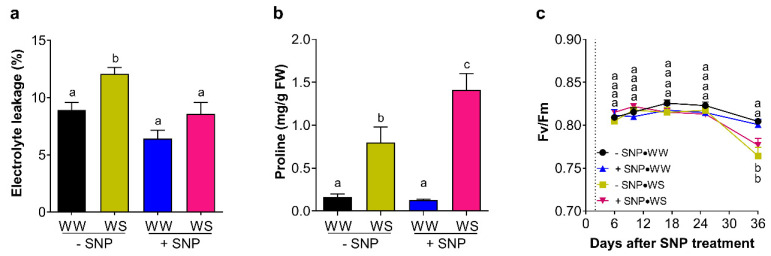
Effect of exogenous NO on electrolyte leakage, proline content and PSII maximum quantum yield in sweet cherry leaves under water restriction. Sweet cherry cv. Lapins plants were treated with 0 mM (−SNP) or 0.5 mM SNP (+SNP). After two days, plants maintained their irrigation (WW) or were exposed to a complete water shortage (WS). (**a**) Electrolyte leakage and (**b**) proline content in leaves sampled at 36 days post SNP treatment. (**c**) Temporal course of the maximum quantum yield of PSII (Fv/Fm) in fully expanded leaves during the water restriction assay. (**a**), n = 6; (**b**), n = 3; (**c**), n = 6. Different letters indicate statistically significant differences in a two-way ANOVA with Tukey’s multiple comparisons test (*p* < 0.05).

## Data Availability

The original contributions presented in the study are included in the article/Appendix A; further inquiries can be directed to the corresponding author.

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
