# Peer review of "Nitric Oxide Mitigates the Deleterious Effects Caused by Infection of Pseudomonas syringae pv. syringae and Modulates the Carbon Assimilation Process in Sweet Cherry under Water Stress"

_plants, 2024, doi:10.3390/plants13101361_

Round 1

Reviewer 1 Report

Comments and Suggestions for Authors

Dear Authors,

In this article, the authors investigate the effect of the addition of a NO donor to cherry plants and its role in water stress and protection against Pseudomonas infection. They have also studied its influence on carbon assimilation and stomatal opening. The results present a large amount of data. However, I believe they should better explain the 'why' behind such a quantity of data, and how it contributes to the initial hypothesis. Introduction and discussion are deficient, it could be improved, and important points are not mentioned. In its current state, the article requires substantial improvement.

Majors:

-The introduction seems quite well, but I believe it's necessary for the authors to enhance it with the following suggestions. At no point do the authors explain how plants synthesize NO. They don't need to delve into detail, but they should indicate the basic pathway of its biosynthesis. I'll provide examples, but they don't have to cite these articles; they can choose others on the topic, such as doi.org/10.3390/ijms231911522.

-In the introduction, the authors dedicate effort to explaining the role of ABA (abscisic acid). It's not clear why they do this, but perhaps the discussion section provides an explanation for this?

-I advise the authors to display the results of the 30 dpt in Figure 3A instead of the 4 dpt. This adjustment will allow the results of the treatment, as shown in Figure 3C, to be more clearly observed.

-Why don't the authors show the results of day 30 dpi t in Figure 3B? This would help to better compare the data with those in Figure 3C.

-L167: “Moreover, disks from 0.2 or 0.5 mM SNP-treated leaves sampled at seven dpt presented necrotic areas of 22 and 20% at four dpt and 26 and 22% at seven dpt, respectively” Poor wording, please rewrite it.

-L196….: It would be advisable for the authors to provide an explanation for why they chose to study those parameters rather than abruptly beginning to show the results.

-L203 “In accordance with these Ψmd values, plants were under moderate water stress in the first date and under a severe water stress in the Second” I'm sorry, I don't understand what the authors mean. Could you please rephrase it?

-Figure 6:  I'm sorry, but I don't understand Figure 6A. What is indicated in the graph is the water loss, and how do they measure it as stated in the figure caption, by weighing the pot? Why do they assume that all the weight loss is solely due to water? And why didn't they do it the same way in the second season?

-In my opinion, for a better explanation, the data related to Figure 7 should be presented before those of Figure 6. This would clarify why the authors analyze the values in Figure 6. I advise the authors to change the order.

-I understand that the results in Figure 7 correspond to the first season. Please indicate this in the figure caption. Ah, I see. They are displaying the results of both seasons in the same figure, as Figure 7C corresponds to the second season. It's important to clarify this. Also, why are the dpt  different in both seasons?

L240: “interact positively” When they say this, what do they mean? Are they suggesting that both effects are synergistic? Has this been documented in other plants? Please clarify this better in the discusión.

L246: “Key physiological parameters” yes but why? Similar to what was mentioned earlier, in this section also, why have they decided to measure stomatal conductance? It would be beneficial to avoid abruptly mentioning results and instead provide context for how these results relate to one another.

-This is an important point. The authors have a lot of results, but sometimes less is more. They should explain why they chose to present those specific data in relation to their initial hypotheses.

-Please provide a better explanation of why you chose to study two seasons, and specifically in Fig 7, what the results of the second season contribute compared to the first. Why is there a big difference between the two seasons?

-L276: “this suggests that with an optimal water status, exogenous NO improves the stomatal conductance and net CO2 assimilation rate with a positive effect on An under water stress” sorry, I doubt about that, based on my understanding of the results in Figure 8, are the data regarding An statistically significant only at 7 dpt in the first season? I don't think this can be claimed as an effect of SNP on An.

L288: As mentioned earlier, why was the decision made to quantify these parameters? What is the rationale behind it? Readers generally appreciate understanding why authors choose to study certain aspects before delving into the results.

Figure 9: Second season? Why not the first too? Please explain Why.

In  the discussion:

-L311: “increasing the antioxidant enzymes” So why haven't they measured the activity of some of these enzymes? Why do they start the discussion by talking about ROS if, out of the large amount of data they have, they don't specifically study that aspect? The discussion is already lengthy enough; please remove these kinds of things. Focus the discussion on the parameters that you have measured, because those will be the important ones, right?

-In the discussion, please, explain the results in an orderly manner. Why are the results of Figure 5 discussed before those of Figure 3?

-L356: “ABA levels in plants do not change, while SA and jasmonic acid increases” I'm sorry, I don't understand the relevance to your study.

-L361: “However, this is likely not the case for plants receiving SNP applications” It is not correct to start a paragraph in this manner; it is unclear what basis is used for this assertion.

-L366: “Therefore, NO biosynthesis and homeostasis can be directly involved in the immune response of sweet cherry plants to bacteria such as Pss”  As mentioned before, describe at least how NO is synthesized, and also, as I mention below, how it degrades into N2O.

-The unused sodium nitroprusside left in the field could remain, causing contamination. Could it be utilized for the production of N2O by bacteria or algae? a significant greenhouse gas. As indicated in 10.3390/ijms23169412. And in respect to that:  I understand that the authors are aiming for these studies to be applied in agriculture, correct? What implications could the accumulation of SNP in the soil have?

-L269-L375: It is not clear why the authors insist on ABA.  As mentioned, the discussion has become overly long. Remove content that only complicates its comprehension.

-L413-428: I insist the part about ABA should be substantially reduced. There are a lot of references that I don't see contributing anything.

-L428-429: On the other hand, while the authors have data on proline, it is barely discussed.

-L489: Bacterial Counting: How do they distinguish if the colonies they count are not contaminations? Do they use antibiotics?

-The discussion needs to be completely reworked. In reality, what the authors present is another summary. And it's better to place it immediately after the discussion, not after materials and methods.

Minors:

-Arabidopsis, Italic please

-Strain [58] Unlike. Typo

-infections [72 ] However, Typo

-L477: . Typo

Reviewer 2 Report

Comments and Suggestions for Authors

In this study sodium nitroprusside was applied to sweet cherry plants cv. Lapins under normal or water-restricted conditions and bacterial canker disease was simultaneously evaluated. In the current scenario of frequent combined stress episodes this study is highly valuable, constructive, and informative. The methodology is appropriate, and data is well analysed with certain key findings. I have minor suggestions.

Abstract: modify lines 51-53 for better clarity

Line 56-57: Is it correct?  “The decreased susceptibility to Pss infection caused either by exogenous NO or water stress”.

Line 73: Please change this line “. It is caused by species and pathovars belonging to the genus…

Please verify that this formula is correct.” The disease severity (DS) per seasonal branch was expressed using the following formula: DS = (number of symptoms in a branch)/3 *100”. Also, how number of symptoms in a branch was calculated?

“After peeling the inoculated zone, the size of the inner injury induced by the infection was measured from the inoculation point to the top of the observed cracking with a hand ruler.” This measurement does not match with parameters in formula. 

Round 2

Reviewer 1 Report

Comments and Suggestions for Authors

Dear Authors,

I think the authors have addressed all my suggestions correctly and I accept the paper in its current version